# GC, GC/MS Analysis, and Biological Effects of Essential Oils from *Thymus mastchina* and *Elettaria cardamomum*

**DOI:** 10.3390/plants11233213

**Published:** 2022-11-23

**Authors:** Nenad L. Vukovic, Milena D. Vukic, Ana D. Obradovic, Milos M. Matic, Lucia Galovičová, Miroslava Kačániová

**Affiliations:** 1Department of Chemistry, Faculty of Science, University of Kragujevac, 34000 Kragujevac, Serbia; 2Department of Biology and Ecology, Faculty of Science, University of Kragujevac, 34000 Kragujevac, Serbia; 3Institute of Horticulture, Faculty of Horticulture and Landscape Engineering, Slovak University of Agriculture, Tr. A. Hlinku 2, 94976 Nitra, Slovakia; 4Department of Bioenergy, Food Technology and Microbiology, Institute of Food Technology and Nutrition, University of Rzeszow, 4 Zelwerowicza Str., 35-601 Rzeszow, Poland

**Keywords:** essential oil, *T. mastichina*, *E. cardamomum*, MDA-MB-468 cell line, antimicrobial effect, antibiofilm activity

## Abstract

Spanish marjoram (*Thymus mastichina*) and cardamom (*Elettaria cardamomum*) are traditional aromatic plants with which several pharmacological properties have been associated. In this study, the volatile composition, antioxidative and antimigratory effects on human breast cancer (MDA-MB-468 cell line), antimicrobial activity, and antibiofilm effect were evaluated. Results obtained via treatment of human breast cancer cells generally indicated an inhibitory effect of both essential oils (EOs) on cell viability (after long-term treatment) and antioxidative potential, as well as the reduction of nitric oxide levels. Antimigratory effects were revealed, suggesting that these EOs could possess significant antimetastatic properties and stop tumor progression and growth. The antimicrobial activities of both EOs were determined using the disc diffusion method and minimal inhibition concentration, while antibiofilm activity was evaluated by means of mass spectrometry. The best antimicrobial effects of *T. mastichina* EO were found against the yeast *Candida glabrata* and the G^+^ bacterium *Listeria monocytogenes* using the disc diffusion and minimal inhibitory concentration methods. *E. cardamomum* EO was found to be most effective against *Pseudomas fluorescens* biofilm using both methods. Similarly, better effects of this oil were observed on G^−^ compared to G^+^ bacterial strains. Our study confirms that *T. mastichina* and *E. cardamomum* EOs act to change the protein structure of older *P. fluorescens* biofilms. The results underline the potential use of these EOs in manufactured products, such as foodstuffs, cosmetics, and pharmaceuticals.

## 1. Introduction

Healthier lifestyles and a healthier environment are new goals of modern civilization. One of the many issues influencing human health is the consumption of less natural and more processed food products. In order to increase the quality, safety, and shelf life of food, industrial processes include the addition of preservatives [1]. The use of synthetic preservatives for this purpose is extensive, but their impact on human life is under discussion [2]. One way of dealing with this issue is the replacement of synthetic preservatives with more effective, low-cost ones obtained from natural sources.

Antibiotics represent one of the most important therapies in dealing with infectious diseases. However, extensive use of antibiotics in community settings, hospitals, and agriculture has led to antibiotic resistance. This growing problem has caused a resurgence in the screening of natural products for medicinal uses [3].

Breast cancer is the second leading cancer-related disease among women. Even though there has been much clinical research, this disease is still the most common cause of cancer death in women worldwide. It has been reported that factors influencing the occurrence of this type of cancer are closely related to estrogens [4,5]. Available treatments such as radiotherapy or chemotherapy cause damage to healthy cells as well as to cancer cells and can increase resistance. This can lead to failure of treatment, so the urgent need to discover alternative ways of treating this disease is currently a focus of investigation. Herbal medicine represents one of many ways of dealing with these limitations. It is well known that plants are rich in bioactive compounds that could serve as bases for chemotherapeutic development.

Plants are known to produce wide ranges of secondary metabolites to ensure the survival of their species. One group of plant metabolic products are volatile or essential oils. Essential oils (EOs) have shown increasing potential in the pharmaceutical, food, and cosmetic industries, as they are widely recognized as safe (by the US FDA (Food and Drug Administration) and the EPA (Environmental Protection Agency)) and are already in use in these industries [6,7,8]. Moreover, many studies have characterized EOs as therapeutics (antioxidant, antimicrobial, anti-allergic, antiviral, enzyme inhibitory, insecticidal, anti-tumor, and pro-apoptotic) depending on their chemical compositions, but further studies are still needed to update the current knowledge base.

Commonly known as Spanish marjoram, *Thymus mastichina* L. is an endemic species that belongs to the Lamiaceae family and usually inhabits the Iberian Peninsula. It is characterized by leaves arranged in opposite pairs and small zygomorphic and bilabiate flowers [9,10]. Traditionally, due to its strong scent of eucalyptus, this plant has been used for treating respiratory, digestive, and rheumatic disorders [11,12]. EOs of this aromatic plant consist of a complex mixture of volatile terpenes and are widely used in the perfume and cosmetics industries [13]. It is well known that the chemical composition of an EO depends on various factors including the plant species, culture, and environmental conditions [9]. Literature data show that the most abundant EO components of this species are 1,8 cineole and linalool, followed by α-pinene, β-pinene, and α-terpineol [10,14]. Previous studies have indicated antibacterial, antifungal, anti-inflammatory, antioxidant, anticancer, antiviral, insecticidal, insect-repellent, and anti-enzymatic (anti-Alzheimer’s, α-amylase, and α-glucosidase) effects of EOs obtained from the aerial parts of this plant, indicating their potential use in the food and pharmaceutical industries [10,13,14,15].

Green or true cardamom (*Elletaria cardamomum* L. Maton), belonging to the Zingiberaceae family, is a plant native to India and Sri Lanka but also cultivated in Guatemala, Nepal, Indonesia, Costa Rica, Mexico, and Tanzania [16,17,18,19]. Plants of this species can grow up to 8-foot-high shrubs with thick, fleshy, lateral roots [20]. Traditionally, it is known as the “Queen of Spices”, since its dried fruit is highly priced as a spice around the world, and in India it is considered the second essential “national spice” [21,22]. Due to its characteristic aroma, this plant is widely used in the food and cosmetics industries as a flavoring and fragrance agent. Moreover, from the pharmaceutical point of view, it is used to treat various disorders, such as gum infections, asthma, bronchitis, nausea, and cataracts, as well as cardiac, digestive, and kidney diseases [16,17,18,19,20,21]. The biological effects of cardamom are in close relationship with their volatile composition. Cardamom EOs are rich in volatiles that have therapeutic benefits, such as ester α-terpinyl acetate, and monoterpenes 1,8-cineole, limonene, linalool, terpinolene, myrcene, and α-pinene, which are responsible for these EOs’ effectiveness in curing different ailments [16,17,18]. Literature data show that EOs obtained from this plant show various biological effects such as antioxidant, antihypertensive, antidiabetic, gastroprotective, laxative, antispasmodic, antibacterial, anti-platelet-aggregation, and anticancer activities [16,17,18,19,23].

Therefore, herein, the chemical composition; effects on cell viability, redox homeostasis, and migratory capacity of the breast cancer MDA-MB-468 cell line; and antibacterial, and antibiofilm activity of *Thymus mastichina* and *Elletaria cardamomum* EOs are investigated, clarified, and discussed compared to results from other previously conducted studies.

## 2. Results

### 2.1. Volatile Composition of T. mastichina Essential Oil

The volatile composition of the analyzed *T. mastichina* EO was determined using GC and GC/MS analysis. The obtained results are presented in Table 1, while Table 2 shows the percentage amounts of each class of identified components. Corresponding chromatograms are presented in Appendix A. In total, 71 volatiles were identified, representing 99.5% of the total. Oxygenated monoterpenes (in the amount of 80.2%) were the dominant components, out of which monoterpene epoxide (47.1%) and monoterpene alcohols (27.8%) were found in the highest amounts. The major constituent of *T. mastichina* essential oil was a monoterpene epoxide (1,8 cineole) at 47.1%, followed by the monoterpene alcohols linalool (19.4%) and α-terpineol (4.6%). Monoterpene hydrocarbons were identified as the second major class of compounds, with α-pinene (3.7%), sabinene (2.1%), and limonene (2.1%) identified as components in high amounts. Other identified classes of compounds were found in amounts less than 1.8%.

### 2.2. Volatile Composition of E. cardamomum Essential Oil

The chemical composition of *E. cardamomum* essential oil, determined via GC and GC/MS analysis, is shown in Table 3, while percentage amounts of each class of identified compounds are presented in Table 4. The obtained chromatograms are shown in Appendix A. Forty-six components in total were identified, which represented 99.5% of the total. The main constituents of the analyzed sample belonged to the class of oxygenated monoterpenes (84.0%) with the monoterpene ester α-terpinyl acetate (37.3%) and the monoterpene epoxide 1,8 cineole (28.6%) as the dominant compounds. Other oxygenated monoterpenes identified in high amounts were linalool acetate (7.9%), linalool (4.5%), and α-terpineol (2.4%). The second major class of identified compounds was monoterpene hydrocarbons at the amount of 13.5%, with sabinene (5.1%), limonene (2.3%), and β-myrcene (2.1%) as the dominant ones. Other identified classes of compounds belonging to the sesquiterpenes and nonterpenic compounds were found in amounts less than 1.3%.

### 2.3. Cell Viability Assays

The next step of this research was to evaluate the effects of the tested EOs on the cell viability of a human breast cancer cell line (MBA-MB-468). During the experiment, cells were treated with six different concentrations (1, 10, 20, 50, 100, and 200 µg/mL) of both tested EOs over short-term (24 h) and long-term (72 h) treatments. The results of the cell viability of MDA-MB-468 cells after 24 and 72 h of exposure to the investigated treatments are presented in Figure 1.

The obtained results showed that all applied treatments of *T. mastichina* EO caused a statistically significant increase in cell viability after 24 h. That effect was pronounced for applied concentrations of 1, 10, and 20 µg/mL. In contrast, applied concentrations of 100 and 200 µg/mL reduced cell viability compared to the control values, but they were not statistically significant. After 72 h of treatment, the viability of cells exposed to all six concentrations of both EOs was decreased compared to control cells, but statistically significant decreases were detected only for *E. cardamomum* essential oil. In addition, cell viability after 72 h of exposure to the examined treatments was further decreased compared to both 24 h and control, indicating time dependence.

### 2.4. The Effects of Essential Oils on Redox Status Parameters

The effects of short-term and long-term exposure of MDA-MB-468 cells to different concentrations of treatment on redox status parameters were monitored, and the obtained results are given in Figure 2 and Figure 3. In order to obtain more comprehensive data on the influences of the examined treatments on the redox homeostasis of breast cancer cells, the concentrations of the following parameters were determined: O_2_^•−^ (nmol/mL in 10^5^ cells/mL), NO (µmol/mL in 10^5^ cells/mL), and total GSH (in µmol/mL in 10^5^ cells/mL).

The data presented in Figure 2 show the concentrations of O_2_^•−^ and NO in MDA-MB-468 cells after 24 and 72 h of exposure to six doses of each EO treatment. The results show that the production of O_2_^•−^ was decreased (statistically significantly) compared to the control after 24 h exposure to all concentrations. After 72 h of treatment, a statistically significant decrease in the production of O_2_^•−^ compared to control cells was also observed, with the most intense drop in production induced by the concentration of 200 µg/mL for both oils.

Regarding the production of NO, the obtained results show that after 24 h of treatment with both EOs, the concentration of NO_2_^−^ (as an indicator of NO) was decreased by all concentrations compared to the control cells. A decrease in NO concentration compared to the control cells was observed in MDA-MB-468 cells even after long-term exposure to all examined concentrations. After long-term treatment, the highest applied concentration (200 µg/mL) provoked the strongest decrease.

The data presented in Figure 3 show the effects of the investigated treatments on the production of total glutathione levels in the breast cancer cells. The level of total GSH after short-term (24 h) exposure of MDA-MB-468 cells to all applied concentrations of EOs was significantly increased compared to control cells. The increase in total GSH level was dose-dependent. The lowest production of GSH was measured in cells treated with 1 µg/mL. After a 72 h incubation period, the same trend in the level of GSH was observed. The decrease in GSH concentration was also proportional to the applied dose of treatment.

### 2.5. Transwell Assay for Cell Migration

To examine the effects of two EO treatments at two concentrations on the migration capacity of MDA-MB-468 cells over long-term treatment, a 2D transwell migration assay was performed. The results indicated a significant dose-dependent decrease in the cell migration index of the cells exposed to both EOs compared to the nontreated cells, as presented in Figure 4. Long-term exposure (72 h) to 10 µg/mL of *T. mastichina* produced the strongest decrease in migration capacity of 33% compared to the control.

### 2.6. Antimicrobial Testing

The antimicrobial activity of *T. mastichina* EO was evaluated using the disc diffusion method (Table 5) and minimal inhibition concentration (Table 6). In this study, we tested seven bacterial and three yeast strains. Using the disc diffusion method, this EO showed the best antimicrobial activity against yeast *C. glabrata* (9.33 ± 1.15 mm) and G^+^ *L. monocytogenes* (9.00 ± 1.00 mm), while the most resistant was G^−^ *E. coli* (5.33 ± 0.58 mm). Overall, the presented results showed weak antimicrobial activity of *T. mastichina* EO. The minimal inhibition concentration (MIC) values ranged from MIC 50 0.86 and MIC 90 2.11 µL/mL to MIC 50 12.35 and MIC 90 15.46 µL/mL. The best antimicrobial effect of *T. mastichina* EO was found against G^−^ *H. influenzae* and *Y. enterocolitica*, as well as against the yeast *C. glabrata* (MIC 50 0.86 and MIC 90 2.11 µL/mL). However, G^+^ *S. pneumoniae*, *S. aureus*, and yeast *C. albicans* were the most resistant to treatment with this EO (MIC 50 12.35 and MIC 90 15.46 µL/mL for all).

Table 6 shows the antimicrobial activity of *E. cardamomum* EO against ten microorganisms (seven bacterial and three yeast strains).

Compared to *T. mastichina* EO, the disc diffusion method (Table 5) showed moderate antimicrobial activity of EO obtained from *E. cardamomum* against five microbial strains (G^−^
*H. influenzae*, *P. fluorescens*, and yeasts *C. glabrata*, and *C. tropicalis*). Antimicrobial activity ranged from 1.00 ± 0.00 to 13.33 ± 1.15 mm, and the most sensitive to treatment with *E. cardamomum* EO was *P. fluorescens* biofilm. Moreover, results obtained using minimal inhibition concentration (Table 6) confirmed that *P. fluorescens* was susceptible to *E. cardamomum* EO (MIC 50 0.86, MIC 90 2.11 µL/mL), but also showed a sensitivity of yeast *C. glabrata* to this EO. On the other hand, G^−^ *E. coli* and G^+^ *S. pneumoniae*, *L. monocytogenas*, and *S. aureus* strains showed resistance to treatment with this EO (MIC 50 49.86 and MIC 90 56.21 µL/mL for all).

### 2.7. Antibiofilm Testing

The effects of *T. mastichina* and *E. cardamomum* EOs against biofilm-producing *P. fluorescens* were evaluated using mass spectrometry over 14 days. This experiment was performed using a MALDI-TOF MS Biotyper in order to analyze the protein spectra of *P. fluorescens* so that changes in the molecular structure of the bacteria following growth inhibition could be observed. The spectra of biofilms and planktonic cells in the control group developed identically and, therefore, the spectra of planktonic cells were used for greater clarity compared to the control spectrum. Experimental treatments consisted of two different surfaces (stainless steel and plastic) treated with both EOs. To make results clearer, control planktonic cells were used to compare the molecular changes of the biofilm.

#### 2.7.1. The Effect of *T. mastichina* EO on *P. fluorescens* Biofilms

Results of the effects of *T. mastichina* EO on the developmental stages of *P. fluorescens* biofilm over 14 days are presented in Figure 5. During days 3 to 9 of the experiment, only slightly lower peak intensity in experimental groups was observed, indicating similarity in the bacterial biofilms of the experimental and control groups (Figure 5A–D). Significant changes in the mass spectra of biofilms on stainless steel and plastic were observed on days 12 and 14 compared to the control sample (Figure 5E,F). No differences in the evolution of the protein profile between the plastic surface and the stainless steel were observed. This finding confirms that *T. mastichina* EO acted to change the protein structure of older biofilms.

A dendrogram constructed according to the mass spectra also confirmed the similarity of the experimental groups of biofilms with the planktonic cells until the 9th day of the experiment. The short MSP distance observed in the dendrogram confirms this conclusion. On the 12th and 14th days of the experiment, there was an increase in the MSP distance of the experimental groups compared to the control groups, indicating that the EO of *T. mastichina* affected the disruption of biofilm homeostasis or contributed to its degradation (Figure 6).

#### 2.7.2. The Effect of *E. cardamomum* EO on *P. fluorescens* Biofilms

Figure 7 shows the spectra of the developmental stages of *P. fluorescens* biofilm throughout the experiment. The mass spectra of young biofilms obtained on the 3rd and 5th days (Figure 7A,B) of cultivation also showed very similar spectra between the control and experimental groups. These results indicate the same protein production in the early conditions. The first spectral changes were noticed on the 7th day of cultivation in the experimental group of *P. fluorescens* biofilm on the stainless steel surface (Figure 7C). Based on the recorded evolution of the spectra, on the 7th day, no significant changes in the protein profile were observed for the experimental group on the plastic surface compared to the control spectra. On day 9, the effect of *E. cardamomum* EO was noted in both experimental groups (Figure 7D). The trend of increasing effect in both experimental groups was maintained until the end of the experiment (Figure 7E,F). Changes were visible in the protein profiles of the *E. cardamomum* EO-treated biofilms, which led to the conclusion that this treatment affected the homeostasis of bacterial biofilms formed on stainless steel and plastic surfaces.

A dendrogram was constructed as a visualization of the mass spectra to determine some similarities in biofilm structure concerning MSP distance. It can be concluded from the constructed dendrogram (Figure 8) that the planktonic stage (P), together with the control groups and young biofilms, had the shortest distance during days 3 and 5 (PFS 3, PFP 3, PFS 5 PFP 5). The similarities in the protein profiles of the control groups were confirmed by short MSP distances. Young biofilms and control planktonic cells also had short MSP distances that matched the mass spectra. The MSP distances of the experimental groups gradually increased over time. Mass spectra analyzed on days 12 and 14 of the experiment had the longest MSP distances, which indicated changes in the molecular profiles of *P. fluorescens* bacteria.

## 3. Discussion

There are many reports in the literature of studies involving EOs obtained from *T. mastichina* and *E. cardamomum*. Previously reported studies of the chemical composition of *T. mastichina* indicate different chemotypes, defined by main compounds present in high amounts: 1,8-cineole, linalool, and 1,8-cineole/linalool [13]. Considering this, our results reveal that the *T. mastichina* EO investigated in this study was of the 1,8-cineole chemotype. Studies on *E. cardamomum* mainly suggest α-terpinyl acetate and 1,8-cineole as major constituents of this EO, which is in accordance with our results [16,24,25]. Additionally, some reports indicate that these two components are responsible for the fragrance of this EO [26]. Previously published studies on EOs of *E. cardamomum* have reported between 11 and 67 compounds identified in total. Slight differences have been noted throughout the literature regarding other compounds present in high amounts. In some studies, 4-terpinen-4-ol and sabinene were detected; in others, linalool, dihydrocarveol, geraniol, *Z*-caryophyllene, *E*-nerolidol, eugenol, and terpinen-4-ol, or sabinene, 4-terpinen-4-ol, and myrcene; in still others, linalool acetate, sabinene, and linalool, in line with our present findings [19,24,27,28]. Reported differences in the chemical profiles of both plant species could affect the observed biological activities of their essential oils, and are observed as a result of different environmental factors influencing plant development as well as the part of the plant used to obtain the EOs.

Medicinal, aromatic plants have been used for centuries for their beneficial properties. Due to their high contents of bioactive secondary metabolites, they have been especially used to treat a variety of human diseases [29]. The current concentrations of conventional chemotherapeutic drugs used in therapies could potentially be reduced if combined with specific doses of EOs, which could also decrease chemotherapy-associated toxicity. However, there is still a lack of preclinical studies of EOs as effective anticancer agents and protective components, requiring further extensive safety and toxicity studies of EOs prior to their applications in clinical trials [30]. EOs have been shown to possess a wide range of anticancer properties and mechanisms of action. Breast cancer is the most common cancer in women globally, with an increasing incidence and persistently high level of mortality [31,32,33]. In attempts to solve this problem, EOs are being considered promising agents for novel anticancer therapies to overcome the side effects and the high cost of chemotherapy approaches in dealing with breast cancer [34]. Considering the number of bioactive components present in EOs, as well as their synergistic actions, it is of high importance to perform further studies regarding content evaluation and the contributions of individual EO components to the overall biological effects of these mixtures. Selective specificity of action against cancer cells is highly required due to a lack of conventional chemotherapeutic strategies [30].

It has been suggested that the preventive effect of EOs against cancer disorders could be related to the promotion of cell cycle arrest, stimulating cell apoptosis and DNA repair mechanisms while inhibiting cancer cell proliferation, metastasis formation, and multidrug resistance. All the above-mentioned factors make EOs potential candidates for supporting anticancer therapeutic agents [35].

Reactive oxygen species produced via various metabolic pathways in the tissues damage the cells by binding to cell components such as carbohydrates, proteins, and DNA. The antioxidant activity of *T. mastichina* has also been widely explored through different assays, representing an interesting alternative to synthetic antioxidants. In some studies, these effects were related to the sample’s composition and tests were conducted to understand whether some compounds were primarily responsible for the observed activity [14].

Results obtained after 24 h and 72 h of incubation with various concentrations of both essential oils generally indicated a slight proliferative effect compared to the nontreated cells after 24 h of incubation with *T. mastichina* EO, but dose-dependent inhibition of cell viability was detected after 72 h treatment with both EOs, indicating considerable antitumor outcomes. The observed antiviability outcomes could be due to the reduced proliferative potential of the tested cells and to the proapoptotic effects of the main constituents of the EOs. Certain studies have also reported the antiproliferative activity of *T. mastichina* EO against human breast carcinoma cell lines, which could be related to 1,8-cineole content [13,36]. However, in the case of complex mixtures, it is more likely that a synergy between various components is responsible for the expressed effects.

Reactive oxygen species (ROS) appear to be involved in the regulation of various physiological pathways, including signal transduction and differentiation. Recently, emerging evidence has suggested the involvement of ROS and the aberrant activation of redox-sensitive signaling pathways in tumor invasion and migration. In our study, significant dose-dependent and time-dependent antioxidative effects were shown in cells treated both EOs compared to the control cells. Some antioxidants may enhance the effects of cytotoxic regimes, improving the response rate of tumors to chemotherapeutic agents, while some others can ameliorate their antitumor activity [37]. Disturbance of oxidative homeostasis is one of the major features of cancer cells in general, so our data indicate that the exerted antioxidant impact could be crucial in the breast cancer cell survival and viability detected in the study.

Nitric oxide (NO) has been suggested to possess both antitumor and protumor properties, depending on tissue type and timing [38]. MDA-MB-468 cells treated with *T. mastichina* EO showed a significant decrease in the production of nitrite, the main indicator of NO concentration, compared to the control after both treatment times. However, *E. cardamomum* EO only induced a significant drop of NO level after long-term treatment. Changes in the production of NO could affect various signaling pathways that involve nitric oxide, leading to potential antitumor outcomes. The recorded NO decrease could be due to transcriptional and post-translational regulation of iNOS enzyme activity, consequently affecting various protumor signal pathways, but could also inhibit angiogenesis and blood supply in the tumor tissue, thus attenuating tumor growth.

The tested EOs’ actions on migration capacity in breast cancer cells were assessed by 2D transwell migration assay. The results indicated a significant dose-dependent decrease in the cell migration index of MDA-MB-468 cells exposed to both EOs compared to the nontreated cells. The inhibitory effects on NO production detected in the study could be one of the mechanisms of antimigratory potential of the tested EOs, since numerous studies indicate that the reduction of nitric oxide levels can inhibit cell migration [39,40].

Previously published studies of the antimicrobial activity of *T. mastichina* EO indicate that a higher concentration of linalool is responsible for better antimicrobial properties [10,13]. Since the EO tested in the present study was from the 1,8-cineole chemotype, lower amounts of linalool could have contributed to weak antimicrobial effects. The observed effect could also be a product of antagonistic and synergistic effects of the various other constituents present in this EO. However, studies conducted by Faleiro et al. showed moderate antimicrobial activity of Algarve *T. mastichina* EOs from the 1,8-cineole chemotype measured by the agar disc diffusion method. In this study, Algarve *T. mastichina* EOs showed the best activity against G^+^ *S. aureus*, showing zone of inhibition diameters of 13.7 mm and 15.7 mm for the flower and leaf essential oils [41]. Using the same method, Ballester-Costa et al. investigated the antimicrobial activity of *T. mastichina* EO of the same chemotype against several bacterial strains. They confirmed that this EO had inhibitory activity against eight bacterial strains, but the highest activity was observed against G^+^ *L. innocua* and G^−^ *A. faecalis* [42]. In the same study, the authors observed better activity of this EO when using a microdilution assay, i.e., the less sensitive bacteria strains in the agar disc diffusion method showed higher susceptibility in the microdilution assay [42]. Cutillas et al. investigated the antimicrobial activities of four different *T. mastichina* EOs using a microdilution assay. Their results also suggested weak inhibition of growth of *E. coli*, *S. aureus*, and *C. albicans* by both chemotypes (linalool and 1,8-cineole), with MIC values from 2.3 mg/mL to 9.4 mg/mL [10].

Literature reports indicate that *E. cardamomum* EO varieties exhibit moderate-to-high activity against selected bacteria and fungi using different antimicrobial bioassays. Tarfaoui et al. investigated the antimicrobial activity of *E. cardamomum* EOs on several bacterial (*S. aureus*, *S. epidermidis*, *E. coli*, *K. pneumoniae*, *P. mirabilis*, *P. aeruginosa*, and *A. baumannii*) and yeast (*C. tropicalis* and *C. albicans*) strains [24]. Using a disc diffusion assay, the best antimicrobial effects among the bacterial strains were observed against *S. aureus* and *S. epidermidis*, with inhibition zones of 20 mm and 14 mm, respectively. The effects of this EO against both yeast strains were similar, showing a zone of inhibition of 13 mm for *C. albicans* and 12 mm for *C. tropicalis*. The results obtained in this study using a microdilution assay only confirmed the findings of disc diffusion assay. The overall conclusion was that G^+^ bacteria are more sensitive to treatment compared to G^−^ bacterial strains [24]. The same conclusion was drawn in the more recent study performed by Al-Zereini et al. [43]. The authors noticed the better antimicrobial activity of *E. cardamomum* EOs against G^+^ *B. subtilis* compared to G^−^ *E. coli* and *E. aerogenes*, which were not inhibited up to the maximum applied concentration in agar diffusion and microbroth dilution assays.

The antibacterial activities of *T. mastichina* and *E. cardamomum* EOs were expressed in terms of zones of inhibition in mm and minimal inhibition concentrations in µL/mL. The results obtained in this study using the disc diffusion method showed a weak antimicrobial effect of the EO from *T. mastichina* against all tested microorganisms. However, the EO from *E. cardomomum* showed moderate antimicrobial activity against four microorganisms and a weak effect against six microorganisms. The EO obtained from *T. mastichina* was the most effective against *C. glabrata*, while *E. cardamomum* showed the best effect against the biofilm-producing bacterium *P. fluorescens*. Additionally, the results obtained in this study, unlike previous findings, showed that the *E. cardamomum* EO displayed better effects against G^−^ bacterial strains compared to G^+^ ones. However, in line with previously published data, G^−^ *E. coli* was resistant to treatment with this EO.

The antibiofilm activities of EOs have been the focus of recent investigations [44,45,46]. It is known that bacterial biofilms can protect bacteria against antimicrobial pressures by taking the form of a physical barrier. Biofilm formation makes bacteria highly resistant to environmental stress compared to planktonic bacteria of the same species [47]. Similarly, it is considered that the planktonic single-celled state is a transitional phase, while the biofilm is in most cases the usual mode of bacterial growth. Various studies have reported abnormalities in protein production associated with biofilm formation and degradation after treatment with EOs [48]. Those abnormalities can be observed using MALDI-TOF mass spectra by comparing untreated samples to treated ones, which was used in this study. Herein, for the first time, we examined the effects of *T. mastichina* and *E. cardamomum* EOs on biofilm-forming *P. fluorescens* bacteria on different surfaces. The results showed changes in the biofilm protein profile of the *P. fluorescens* after treatment with these two EOs. It can be concluded that *T. mastichina* and *E. cardamomum* EOs affected the homeostasis of bacterial biofilms formed on stainless steel and plastic surfaces. Previous studies revealed that treatment with *T. mastichina* EO prevented the formation of biofilms by *P. aeruginosa* and *S. aureus* [49]. Additionally, E. cardamomum EO showed high antibiofilm potential against biofilms formed by *E. coli* and *B. subtilis* [25]. In another study, this EO inhibited biofilm formation inhibition by *E. coli* and *S. typhimurium* [50]. Even though many studies have shown an inhibitory activity of EOs on the formation of biofilms, the mechanism behind their antibiofilm activity is generally not understood well to date. The current predictions suggest that the inhibition of some enzymes that are involved in the formation of biofilm could be responsible for EOs’ activity [25].

## 4. Materials and Methods

### 4.1. Essential Oils

*Thymus mastichina* and *Elettaria cardamomum* EOs were purchased from Hanus, s.r.o. (Nitra, Slovakia) and were prepared by steam distillation of dried flowering stalks. They were stored in the dark at 4 °C throughout the analysis.

### 4.2. Gas Chromatography–Mass Spectrometry and Gas Chromatography Analyses of T. mastichina and E. cardamomum

Identification of volatile constituents in EO samples was performed on an Agilent Technologies (Palo Alto, Santa Clara, CA, USA) 6890 N gas chromatograph. The chromatograph was equipped with a quadrupole mass spectrometer 5975 B (Agilent Technologies, Santa Clara, CA, USA), using an HP-5MS capillary column (30 m × 0.25 mm × 0.25 µm). The chromatograph was interfaced and operated by HP Enhanced ChemStation software (Agilent Technologies). The injection volume of the EO sample diluted in hexane (10% solution) was 1 µL. The carrier gas used was helium 5.0, with a flow rate of 1 mL/min. Split/splitless injector temperature was set at 280 °C, MS source and MS quadruple temperature were set at 230 °C and 150 °C, respectively, and the mass scan range was 35–550 amu at 70 eV. The solvent delay time was 3.20 min for EO sample analysis. For n-alkanes (C7–C35), solvent delay time was 2.30 min in order to obtain the retention index for n-heptane.

GC and GC-MS analysis of the *T. mastichina* sample for Kovats retention index calculations was done under the following chromatographic conditions: temperature program of 50 °C to 90 °C (rate of increase 3 °C/min), held 2 min at 90 °C, 90 °C to 130 °C (rate of increase 4 °C/min), and 130 °C to 290 °C (rate of increase 5 °C/min); the total run time was 57 min, and split ratio was 40.8:1. For the purpose of experimental determination of Van den Dool retention indices, some minor changes were made to the chromatographic conditions. The temperature program was 60 °C to 260 °C with an increasing rate of 3 °C/min, the total run time was 67 min, and the split ratio was 20:1.

The chromatographic conditions for GC and GC-MS analysis of the *E. cardamomum* sample for Kovats retention index calculations were as follows: temperature program of 50 °C to 70 °C (rate of increase 3 °C/min), held 3 min at 70 °C, 70 °C to 120 °C (rate of increase 4 °C/min), and 120 °C to 290 °C (rate of increase 5 °C/min); total run time was 56 min and split ratio was 40.8:1. Analysis of Van den Dool retention indices included the following chromatographic conditions: temperature program of 60 °C to 260 °C with a rate of increase of 3 °C/min, total run time of 67 min and split ratio of 20:1.

The volatile components of the analyzed EOs were identified by comparison of their retention indices (RI) as well as the reference spectra reported in the literature and the ones stored in the MS library (Wiley7Nist) [51,52]. Semiquantification of the components was performed via GC-FID using the same HP-5MS capillary column, taking into consideration amounts higher than 0.1%.

### 4.3. Cell Culture and Treatment

The human breast cancer cell line MDA-MB-468 was obtained from the American Tissue Culture Collection. The cells were cultivated in DMEM with 10% FBS and an antibiotic mixture (100 IU/mL penicillin and 100 µg/mL streptomycin) to the confluence of 70 to 80%. The cells were placed in a 96-well microplate (10,000 cells per well) and propagated in a humidified atmosphere with 5% CO_2_ at 37 °C. After 24 h of incubation, 100 μL measures of medium containing different concentrations of EOs of *T. mastichina* and *E. cardamomum* (1 µg/mL to 200 µg/mL) were added and the cells were incubated for 24 h and 72 h, after which the measurements of cell viability, redox balance parameters, and migration capacity were conducted. Nontreated cells were used as a control, and all treatment concentrations were obtained by serial dilutions of stock solution. All concentrations were applied in triplicate for all the methods.

### 4.4. Determination of Cell Viability (MTT Assay)

The viability of the cells was determined using an MTT assay [53]. The cells were plated at a density of 100,000 cells/mL (100 µL/well) in 96-well plates with DMEM. After an incubation period of 24 h at a temperature of 37 °C in a 5% CO_2_ atmosphere, six different concentrations of EOs ranging from 1 to 200 µg/mL were added to the wells at 100 µL volume per well. The untreated cells served as a control. Both treated and control cells were incubated for 24 and 72 h, after which the cell viability was determined via MTT assay. After a period of incubation, 20 µL of MTT (concentration of 5 mg/mL) was added to each well. MTT is a yellow tetrazolium salt that is reduced to purple formazan in the presence of mitochondrial dehydrogenase. During this reaction, which started after approximately three hours, the formed crystals were dissolved in 20 µL of DMSO. The color formed in the reaction was measured using an ELISA reader at a wavelength of 550 nm. The percentage of viable cells was calculated as the ratio between the absorbance at each dose of the treatment and the absorbance of the nontreated control multiplied by 100 to give a percentage.

### 4.5. Measurement of Superoxide Anion Radical (NBT Test)

The concentrations of superoxide anion radical (O_2_^•−^) in the samples were determined via the spectrophotometric method, which is based on the reduction of nitroblue tetrazolium (NBT) to nitroblue-formazan in the presence of O_2_^•−^ [54]. The assay was performed by adding 20 μL of 5 mg/mL NBT to each well, followed by cell incubation for 1 h at 37 °C in 5% CO_2_. To quantify the formazan production, formazan was solubilized in 20 μL DMSO. The absorbances were measured using an ELISA microplate reader at 550 nm. The concentrations of O_2_^•−^ were expressed as nanomoles per milliliter (nmoL NBT/mL) in 10^5^ cells/mL.

### 4.6. Measurement of NO Concentration (Griess Method)

The spectrophotometric determination of nitrites (NO_2_^−^) as an indicator of the nitric oxide (NO) level was performed using the Griess method [55]. The concentration of NO_2_^−^ is directly proportional to the intensity of the purple color measured by an ELISA reader at 550 nm. THe Griess reaction is based on the coupling of NO-generated diazonium ion with N-(1-napthyl) ethylenediamine, wherein a chromophoric product is formed. Equal volumes of 0.1% (1 mg/mL)*N*-1-napthylethylenediamine dihydrochloride and 1% (10 mg/mL) sulfanilamide solution in 5% phosphoric acid were mixed to form the Griess reagent immediately prior to application to the plate. Over 10 min of incubation (at room temperature, protected from the light sources), a purple color developed. After incubation, absorbances were measured and the nitrite concentrations were expressed in μmoL NO_2_^−^/mL in 10^5^ cells/mL.

### 4.7. Total Glutathione Concentration

The determination of total glutathione was based on the oxidation of reduced glutathione using the DTNB reagent to form a yellow TNB product [56]. The method was performed on cells seeded in a microtiter plate (10,000 cells in 100 μL). After a period of incubation with treatment, the supernatant was removed, 150 μL of 2.5% sulfosalicylic acid was added, and the plate was sonicated. After sonication, 50 μL of supernatant was reacted with 50 μL of the reaction mixture (1 mM DTNB, 1 mM NADPH, and 0.7 U glutathione reductase in 100 mM phosphate buffer). After 5 min of incubation in the dark at room temperature, absorbances were read using an ELISA reader, and the concentration of total glutathione was expressed as μmoL/mL in 10^5^ cells/mL.

### 4.8. Transwell Assay for Cell Migration

The cell migration capacity was determined by the ability of cells to pass through the pores of polycarbonate membranes (pore size 8 µm; Greiner Bio-One, Gallen, Switzerland) at the bottom of transwell chambers. The migration test was performed according to the protocol described by Chen [56]. The cells were exposed to 1 µg/mL and 10 µg/mL concentrations of both essential oils for 72 h. The control cells were cultured in DMEM only. After the treatment exposures, all groups of treated cells were trypsinized and placed in the upper chambers at a density of 100,000 cells/well in 500 µL of DMEM with 10% FBS. The lower chambers of the control cells contained 750 µL of DMEM supplemented with 10% FBS, whereas the lower chambers with treated cells were filled with 1 µg/mL and 10 µg/mL concentration of both treatments. After 6 h of incubation at 37 °C, the cells from the upper surface of the filter were completely removed with gentle swabbing. The remaining migrated cells were fixed for 20 min at room temperature in 4% paraformaldehyde and stained with 0.1% crystal violet in 200 mM 2-(N-Morpholino) ethanesulfonic acid (pH 6.0) for 10 min. Next, 10% acetic acid was used to dissolve the dye and the absorbance was measured at 595 nm. The migration index was calculated as the ratio of absorbance of the treated samples divided by the absorbance of the nontreated control cell value and multiplied by 100 to give a percentage.

### 4.9. Tested Microorganisms

Microorganisms (Escherichia coli CCM 3954, Haemophilus influenzae CCM 4454, Streptococcus pneumoniae CCM 4501, Listeria monocytogenes CCM 4699, Staphylococcus aureus subsp. aureus CCM 2461, Yersinia enterocolitica CCM 7204, Candida glabrata CCM 8270, Candida tropicalis CCM 8264, Candida albicans CCM 8261) were obtained from the Czech Microorganism Collection. The biofilm-forming bacterium Pseudomonas fluorescens was obtained from fish. Identification of Pseudomonas fluorescens was performed using 16 S rRNA sequencing and a MALDI-TOF MS Biotyper.

### 4.10. Antimicrobial Activity—Disc Diffusion Method

The susceptibility of a bacterial strain to treatment with the *T. mastichina* and *E. cardamomum* EOs was determined using the disc diffusion method. The microbial inocula were cultivated over a period of 24 h on Tryptone soya agar (TSA, Oxoid, Basingstoke, UK) at 37 °C for bacteria, and on Sabouraud dextrose agar (SDA, Oxoid, Basingstoke, UK) at 25 °C for yeasts. The inoculum density was set at 0.5 McFarland density standard (1.5 × 10^8^ CFU/mL). A volume of 100 μL of inoculum was added to plates with Mueller Hinton agar (MHA, Oxoid, Basingstoke, UK). Sterile discs with a diameter of 6 mm were saturated with 10 μL of *T. mastichina* and *E. cardamomum* EOs. Discs prepared in this way were placed on the layer of agar with a microbial suspension. Incubation of samples lasted for 24 h at a temperature of 37 °C for bacteria and 25 °C for yeasts. Two antibiotics (cefoxitin, gentamicin, Oxoid, Basingstoke, UK) were used as positive controls for G^+^ and G^−^ bacterial strains. For yeast, the antifungal fluconazole (Oxoid, Basingstoke, UK) was used as a positive control. A disc impregnated with 0.1% DMSO (dimethylsulfoxide, Centralchem, Bratislava, Slovak) served as the negative control. After 24 h incubation, the radius of the inhibition zone (from the edge of the disc to the edge of the zone) was measured. The experiment was performed in three repetitions and the average inhibition zone was calculated [57].

Criteria for detection of inhibitory activity were as follows: an inhibition zone diameter above 5 mm—weak inhibitory activity, above 10 mm—moderate inhibition, and above 15 mm—very strong inhibition. Each test was done in triplicate.

### 4.11. Minimum Inhibitory Concentration (MIC)

In order to determine MIC values, the microorganisms were aerobically cultured. Cultivation was performed in Mueller Hinton Broth (MHB, Oxoid, Basingstoke, UK) at 37 °C for bacteria and in Sabouraud dextrose broth (SDB, Oxoid, Basingstoke, UK) at 25 °C for yeasts over a period of 24 h. The microbial suspension was applied in a 96-well microtiter plate at a volume of 50 µL (optical density of 0.5 McFarland standard). MHB (100 μL) containing EO in a concentration range of 400 μL/mL to 0.2 μL/mL was added to the sample. The concentration ranges were obtained by serial dilution. The contents of each well were thoroughly mixed by pipetting. MHB and SDB with inocula were used as positive controls. The negative controls were MHB, SDB, and EOs [58].

### 4.12. Analysis of Differences in Biofilm Development with MALDI-TOF MS Biotyper

The representative G^−^ biofilm-forming bacterium *P. fluorescens*, obtained from fish, was used in this experiment. Using a MALDI-TOF MS Biotyper, the various phases of change in the protein structure of the biofilms developing on plastic and stainless steel surfaces were estimated after treatment with the tested EOs. The samples (experimental and control) were prepared in polypropylene tubes (50 mL) using 20 mL of MHB and a plastic and a stainless steel slide. The experimental groups used were MHB enriched with 0.5% of the EOs. The inoculated experimental groups were incubated at 37 °C on a 45° slope shaker at 170 rpm. Before analysis, samples of biofilm were taken using a cotton swab from the plastic and stainless steel slides. Analysis was performed by imprinting samples onto a MALDI-TOF metal target plate. Samples of biofilm and planktonic cells were analyzed on the 3rd, 5th, 7th, 9th, 12th, and 14th days. The planktonic cells were obtained by removing 300 µL of culture medium, which was centrifuged at 12,000 rpm for 1 min. The obtained supernatant was removed, and the planktonic cells were washed three times using ultrapure water. Washed cells were applied to a target plate in a suspension volume of 1 μL. Next, 1 μL of α-cyano-4-hydroxycinnamic acid matrix (10 mg/mL) was applied to the biofilm and planktonic cell samples and dried at room temperature. Using a MALDI-TOF MicroFlex (Bruker Daltonics, Billerica, MA, USA), the samples were processed in the range of *m*/*z* 200–2000 after crystallization (in linear and positive mode). Spectral data were obtained via automatic analysis. The same sample similarities were used to generate a standard global spectrum (MSP). The dendrograms were obtained using Euclidean distance. For this purpose, 19 MSP from the spectra generated by the MALDI-TOF Biotyper 3.0 were grouped [59].

### 4.13. Statistical Analyses

All data were evaluated using IBM-SPSS 23 software for Windows (SPSS Inc., Chicago, IL, USA). The data were presented as a mean ± standard error (S.E.M). The statistical significance was determined using a paired-sample T test. The level of statistical significance was set at *p* < 0.05. One-way analysis of variance (ANOVA) was performed using Prism 8.0.1 (GraphPad Software, San Diego, CA, USA), followed by Tukey’s test at *p* < 0.05.

## 5. Conclusions

In conclusion, attending to their traditional uses and reported biological activities, *T. mastichina* and *E. cardamomum* EOs could play noteworthy roles as preservatives in the food industry, and as sources of bioactive compounds in the pharmaceutical industry.

The examined EOs showed desirable outcomes regarding certain features of breast cancer cells important for tumor development. The EOs exerted an overall inhibitory effect on cell viability and showed antioxidative potential, and also reduced the levels of nitric oxide. Similarly, the effects of these oils on the migration capacity of MDA-MB-468 cells suggest that the tested EOs could exhibit significant antimetastatic properties and stop tumor progression and growth. These data indicate that these oils could be interesting and promising agents for further investigation with the aim of advancing the present antitumor chemotherapeutic strategies.

The results of the studies of antimicrobial effects showed that *E. caradamomum* EO was more effective against the tested microorganisms than *T. mastichina*, which displayed only a weak antimicrobial effect. The yeast *C. glabrata* was the most sensitive to both EOs in this study. *E. caradamomum* EO also showed better activity towards G^−^ bacterial strains compared to G^+^ ones. Our study also determined that both EOs degraded old biofilms.

## Figures and Tables

**Figure 1 plants-11-03213-f001:**
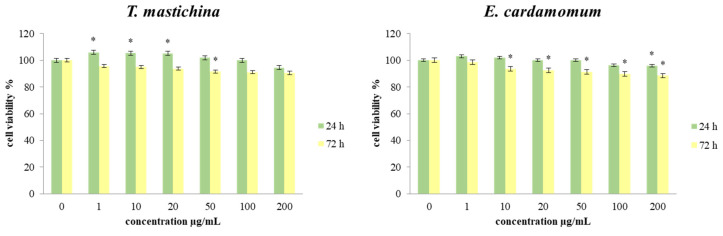
The effects of control treatment and six concentrations of EOs on MDA-MB-468 cell viability after 24 and 72 h of treatment. Results are presented as the mean of three independent experiments ± standard error: * *p* < 0.05 relative to control.

**Figure 2 plants-11-03213-f002:**
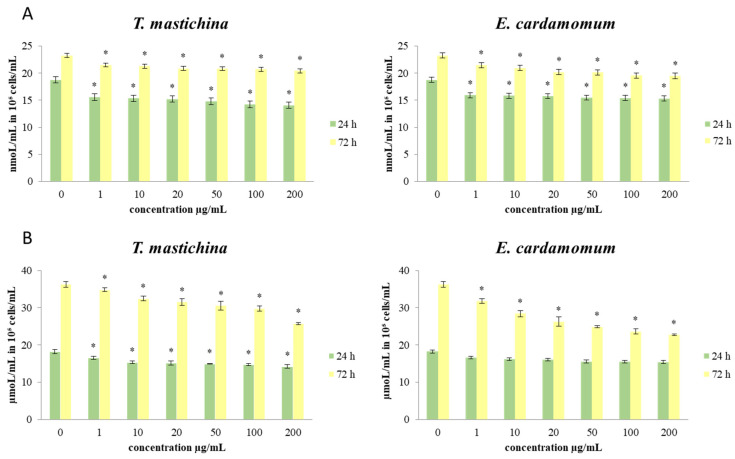
The effects of six concentrations of essential oils on the concentrations of (**A**) O_2_^•−^ and (**B**) NO_2_^−^ in MDA-MB-468 cells after 24 and 72 h of treatment. Results are presented as the mean of three independent experiments ± standard error; * *p* < 0.05 relative to control.

**Figure 3 plants-11-03213-f003:**
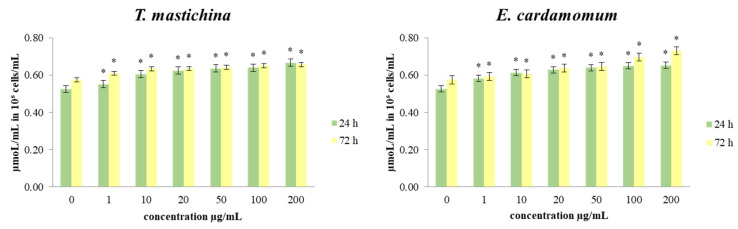
The effects of six concentrations of essential oils on the concentration of totGSH in MDA-MB-468 cells after 24 and 72 h of treatment. Results are presented as the mean of three independent experiments ± standard error; * *p* < 0.05 relative to control.

**Figure 4 plants-11-03213-f004:**
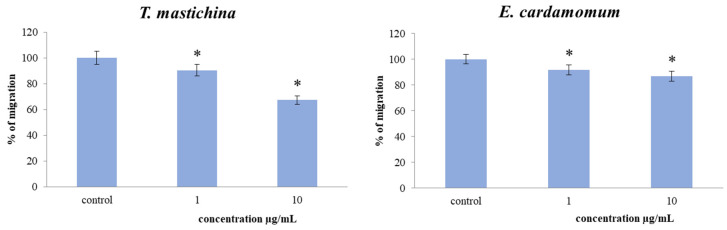
Effect of exposure to investigated essential oils on migration index of MDA-MB-468 cells. The cells were treated at concentrations of 1 µg/mL and 10 µg/mL over 72 h exposure and compared to nontreated control cells. Results are presented as the mean of three independent experiments ± standard error; * *p* < 0.05 relative to control.

**Figure 5 plants-11-03213-f005:**
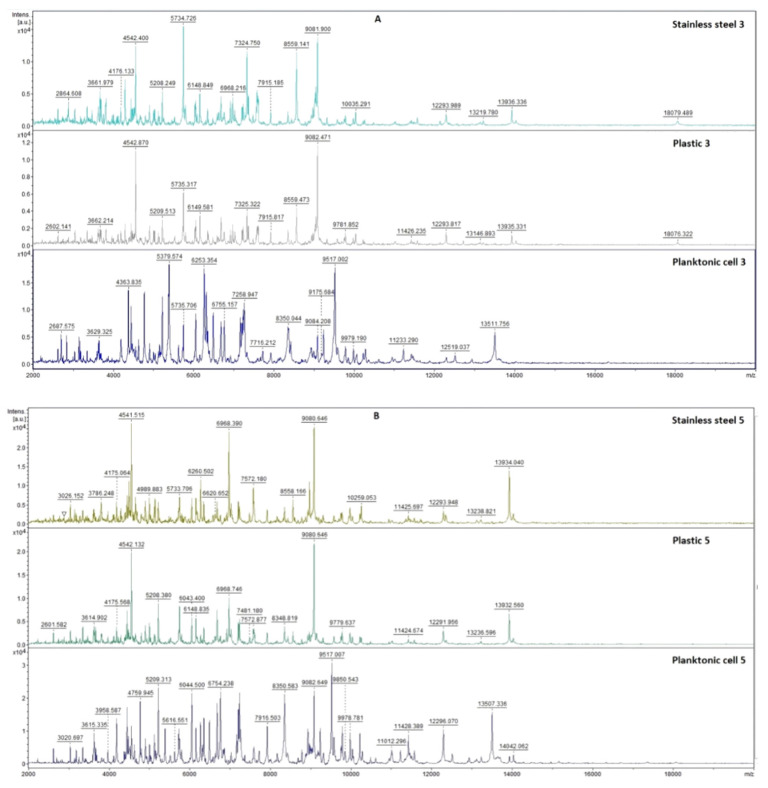
MALDI-TOF mass spectra of *P. fluorescens* during biofilm development: (**A**) 3rd day, (**B**) 5th day, (**C**) 7th day, (**D**) 9th day, (**E**) 12th day, and (**F**) 14th day.

**Figure 6 plants-11-03213-f006:**
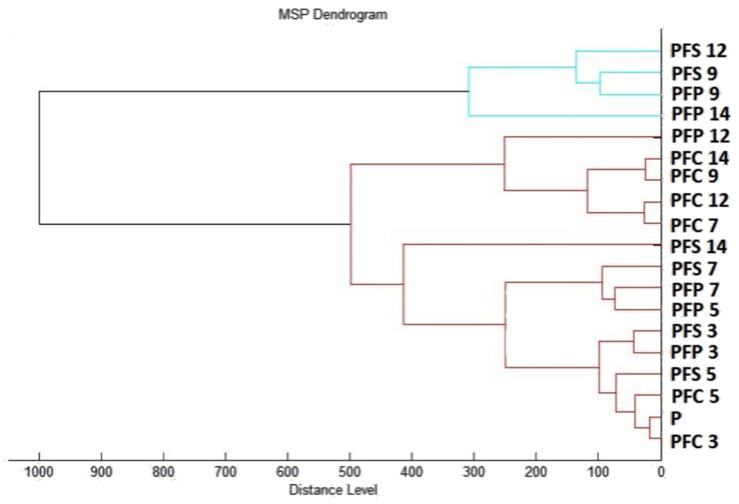
A dendrogram of *P. fluorescens* was generated using MSPs of the planktonic cells and the control. PF, *P. fluorescens*; C, control; S, stainless steel; P, plastic; and only P, planktonic cells.

**Figure 7 plants-11-03213-f007:**
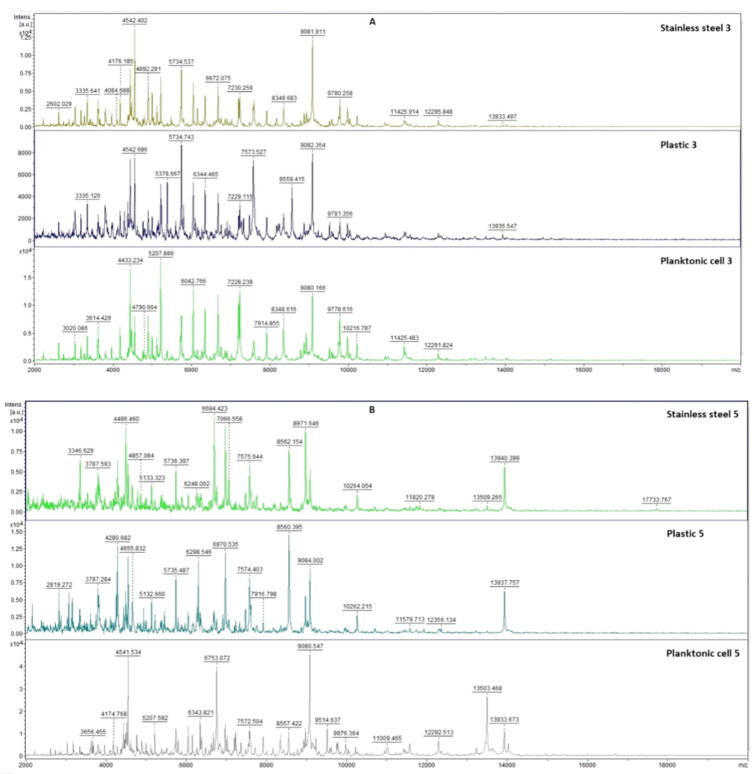
MALDI-TOF mass spectra of *P. fluorescens* during the development of biofilm: (**A**) 3rd day, (**B**) 5th day, (**C**) 7th day, (**D**) 9th day, (**E**) 12th day, and (**F**) 14th day.

**Figure 8 plants-11-03213-f008:**
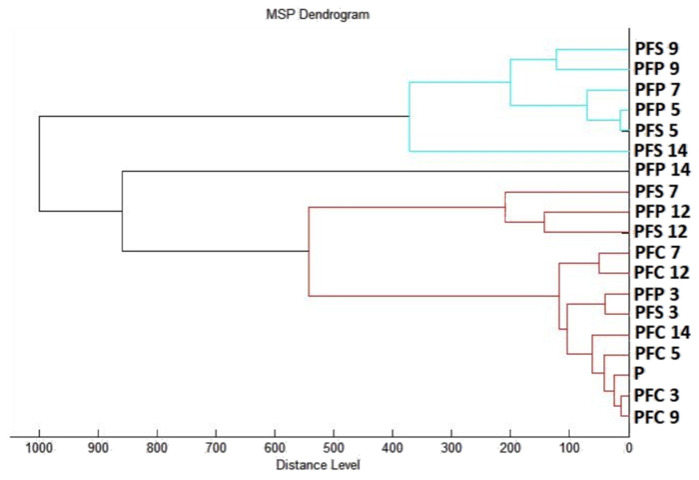
A dendrogram of *P. fluorescens* was generated using the MSPs of planktonic cells and the control. PF, *P. fluorescens*; C, control; S, stainless steel; P, plastic; and only P, planktonic cells.

**Table 1 plants-11-03213-t001:** Volatile composition of essential oil of *T. mastichina*.

No	Compound ^b^	%	AI ^a^		KI ^a^	
			(lit.)	(calc.)	(lit.)	(calc.)
1	tricyclene	tr ^c^	912	922	926	925
2	α-thujene	0.1	924	924	930	928
3	α-pinene	3.7	932	932	939	937
4	camphene	0.6	946	949	954	953
5	verbenene	tr	961	952	967	957
6	sabinene	2.1	969	971	975	975
7	β-pinene	4	974	976	979	980
8	3-octanone	tr	979	983	983	986
9	β-myrcene	1.5	988	987	990	990
10	dehydro-1,8-cineole	tr	988	990	991	991
11	3-octanol	tr	988	998	991	997
12	δ-2-carene	tr	1001	1000	1002	999
13	α-phellandrene	0.1	1002	1007	1002	1007
14	δ-3-carene	tr	1008	1009	1011	1009
15	α-terpinene	0.2	1014	1016	1017	1019
16	*p*-cymene	0.8	1020	1024	1024	1027
17	limonene	2.1	1024	1028	1029	1031
18	1,8-cineole	47.1	1026	1037	1031	1038
19	(*E*)-β-ocimene	0.7	1044	1044	1050	1049
20	γ-terpinene	0.4	1054	1056	1059	1060
21	cis-linalool oxide	0.3	1067	1069	1072	1072
22	α-terpinolene	0.1	1086	1084	1088	1085
23	trans-linalool oxide	0.2	1084	1086	1086	1087
24	*p*-cymenene	tr	1089	1089	1091	1090
25	linalool	19.4	1095	1104	1096	1104
26	hotrienol	0.2	e	1105	1109 ^d^	1106
27	cis-*p*-menth-2-en-1-ol	tr	1118	1124	1137	1141
28	nopinone	tr	1135	1138	1140	1142
29	trans-pinocarveol	0.1	1135	1140	1139	1145
30	camphor	0.7	1141	1147	1146	1151
31	pinocarvone	tr	1160	1162	1164	1166
32	δ-terpineol	1.2	1162	1170	1169	1174
33	borneol	0.7	1165	1172	1169	1175
34	4-terpineol	1.2	1174	1180	1177	1182
35	*p*-cymen-8-ol	tr	1179	1187	1182	1188
36	α-terpineol	4.6	1186	1196	1188	1196
37	verbenone	tr	1204	1207	1205	1205
38	trans-carveol	tr	1215	1218	1216	1219
39	nerol	tr	1227	1223	1229	1225
40	citronellol	tr	1223	1225	1225	1229
41	isobornyl formate	0.1	1235	1227	1239	1229
42	carvone	tr	1239	1242	1243	1245
43	linalool acetate	1.8	1254	1249	1257	1253
44	geranial	tr	1264	1266	1267	1270
45	bornyl acetate	0.3	1284	1283	1285	1284
46	thymol	0.1	1289	1289	1290	1290
47	δ-terpinyl acetate	0.3	1316	1311	1317	1313
48	α-terpinyl acetate	1.8	1346	1345	1349	1348
49	neryl acetate	0.1	1359	1357	1361	1360
50	α-ylangene (α-copaene)	tr	1373	1373	1375	1376
51	geranyl acetate	0.2	1379	1377	1381	1379
52	β-bourbonene	tr	1387	1381	1388	1383
53	β-elemene	tr	1389	1387	1390	1389
54	α-gurjunene	tr	1409	1404	1409	1405
55	trans-caryophyllene	1.1	1417	1417	1419	1419
56	α-trans-bergamotene	tr	1432	1430	1434	1433
57	aromadendrene	0.1	1439	1435	1441	1439
58	α-humulene	tr	1452	1452	1454	1456
59	germacrene D	0.1	1480	1478	1481	1481
60	viridiflorene	0.1	1496	1489	1496	1492
61	bicyclogermacrene	0.2	1500	1492	1500	1494
62	α-muurolene	tr	1500	1495	1500	1497
63	δ-amorphene	0.1	1511	1509	1512	1512
64	γ-cadinene	0.1	1513	1515	1513	1518
65	α-cadinene	tr	1537	1533	1538	1537
66	elemol	0.2	1548	1545	1549	1549
67	spathulenol	0.1	1577	1573	1578	1577
68	caryophyllene oxide	0.1	1581	1578	1582	1583
69	veridiflorol	0.2	1592	1590	1592	1594
70	γ-eudesmol	0.1	1630	1628	1632	1628
71	α-eudesmol	0.2	1651	1651	1653	1648
	total	99.5				

^a^ Values of retention indices on HP-5MS column; ^b^ identified compounds; ^c^ tr—compounds identified in amounts less than 0.1%; ^d^ literature value; ^e^ not found in literature data.

**Table 2 plants-11-03213-t002:** The total amounts of volatiles, presented in percentages for each class of compounds.

Class of Compounds	%
monoterpenes	96.6
*monoterpene hydrocarbons*	16.4
*oxygenated monoterpenes*	80.2
monoterpene epoxide	47.1
monoterpene alcohols	27.8
monoterpene ketones	0.7
monoterpene esters	4.6
sesquiterpenes	2.7
*sesquiterpene hydrocarbons*	1.8
*oxygenated sesquiterpenes*	0.9
sesquiterpene alcohols	0.8
sesquiterpene epoxides	0.1
nonterpenic compounds	0.2
*ketones*	tr
*alcohols*	0.2
total	99.5

**Table 3 plants-11-03213-t003:** Volatile composition of essential oil of *E. cardamomum*.

No	Compound ^b^	%	AI ^a^		KI ^a^	
			(lit.)	(calc.)	(lit.)	(calc.)
1	α-thujene	0.3	924	923	930	926
2	α-pinene	1.9	932	931	939	933
3	α-fenchene	tr ^c^	945	946	952	947
4	camphene	tr	946	948	954	949
5	sabinene	5.1	969	971	975	973
6	β-pinene	0.5	974	977	979	977
7	6-methyl-5-hepten-2-one	tr	981	981	985	985
8	β-myrcene	2.1	988	987	990	990
9	octanal	tr	998	1003	998	1004
10	α-phellandrene	tr	1002	1006	1002	1007
11	α-terpinene	0.1	1014	1016	1017	1020
12	*p*-cymene	0.3	1020	1023	1024	1029
13	Limonene	2.3	1024	1028	1029	1034
14	1,8-cineole	28.6	1026	1034	1031	1041
15	(*E*)-β-ocimene	tr	1044	1043	1050	1052
16	γ-terpinene	0.8	1054	1056	1059	1063
17	cis-sabinene hydrate	0.7	1065	1069	1070	1074
18	α-terpinolene	0.1	1086	1084	1088	1087
19	linalool	4.5	1095	1100	1096	1101
20	(*E*)-4,8-dimethyl-1,3,7-nonatriene	tr	e	1112	1110 ^d^	1114
21	cis-*p*-menth-2-en-1-ol	tr	1118	1123	1121	1127
22	cis-limonene oxide	tr	1132	1131	1136	1135
23	trans-limonene oxide	tr	1137	1135	1142	1140
24	δ-terpineol	tr	1162	1169	1166	1173
25	borneol	tr	1165	1171	1169	1174
26	4-terpineol	1.1	1174	1180	1177	1182
27	*p*-cymen-8-ol	tr	1179	1186	1182	1188
28	α-terpineol	2.4	1186	1195	1188	1196
29	decanal	tr	1201	1206	1201	1207
30	octanol acetate	0.1	1211	1209	1213	1211
31	neral	0.2	1235	1237	1238	1240
32	linalool acetate	7.9	1254	1250	1257	1254
33	geranial	0.3	1267	1266	1264	1270
34	bornyl acetate	tr	1284	1283	1285	1285
35	δ-terpinyl acetate	tr	1316	1313	1317	1313
36	methyl geranate	0.1	1322	1320	1324	1322
37	α-terpinyl acetate	37.3	1346	1352	1349	1355
38	α-ylangene	tr	1373	1368	1375	1372
39	geranyl acetate	0.9	1379	1377	1381	1379
40	β-elemene	tr	1389	1387	1390	1390
41	aromadendrene	tr	1439	1446	1441	1451
42	germacrene D	tr	1480	1478	1481	1482
43	β-selinene	0.3	1489	1486	1490	1490
44	α-selinene	0.1	1498	1492	1498	1496
45	γ-cadinene	0.2	1513	1510	1513	1514
46	(*E*)-nerolidol	1.3	1561	1559	1563	1562
	Total	99.5				

^a^ Values of retention indices on HP-5MS column; ^b^ identified compounds; ^c^ tr—compounds identified in amounts less than 0.1%; ^d^ literature value; ^e^ not found in literature data.

**Table 4 plants-11-03213-t004:** The total amounts of volatiles, presented in percentages for each class of compounds.

Class of Compounds	%
monoterpenes	97.5
*monoterpene hydrocarbons*	13.5
*oxygenated monoterpenes*	84.0
monoterpene epoxide	28.6
monoterpene alcohols	9.0
monoterpene aldehydes	0.2
monoterpene esters	46.2
sesquiterpenes	1.9
*sesquiterpene hydrocarbons*	0.6
*oxygenated sesquiterpenes*	1.3
sesquiterpene alcohols	1.3
nonterpenic compounds	0.1
*ketones*	tr
*aldehydes*	tr
*esters*	0.1
total	99.5

**Table 5 plants-11-03213-t005:** Disc diffusion method.

Microorganism	Zone of Inhibition (mm)	Activity of EO	Zone of Inhibition (mm)	Activity of EO
	*T. mastichina EO*		*E. cardamomum EO*	
*Escherichia coli* CCM 3954	5.33 ± 0.58 ^a,b^	*	1.33 ± 0.58 ^b,a^	-
*Haemophilus influenzae* CCM 4454	8.00 ± 1.00 ^a,b^	*	10.33 ± 0.58 ^b,a^	**
*Streptococcus pneumoniae* CCM 4501	6.33 ± 0.58 ^a,b^	*	1.00 ± 0.00 ^b,a^	-
*Listeria monocytogenes* CCM 4699	9.00 ± 1.00 ^a,b^	*	1.00 ± 0.00 ^b,a^	-
*Staphylococcus aureus* CCM 2461	6.33 ± 1.53 ^a,b^	*	1.67 ± 0.58 ^b,a^	-
*Yersinia enterocolitica* CCM 7204	5.67 ± 0.58 ^a,b^	*	10.00 ± 0.00 ^b,a^	*
*Pseudomonas fluorescens*-biofilm	5.67 ± 0.58 ^a,b^	*	13.33 ± 1.15 ^b,a^	**
*Candida glabrata* CCM 8270	9.33 ± 1.15 ^a,b^	*	11.33 ± 1.53 ^b,a^	**
*Candida tropicalis* CCM 8264	8.33 ± 1.53 ^a,b^	*	11.00 ± 1.00 ^b,a^	**
*Candida albicans* CCM 8261	6.33 ± 0.58 ^a,b^	*	8.00 ± 1.00 ^b,a^	*

Letters (a,b) in lower case indicate a statistical difference *p* ≤ 0.05 between the essential oils within a single microorganism. * Weak antimicrobial activity, ** moderate inhibitory activity.

**Table 6 plants-11-03213-t006:** Minimal inhibition concentration.

Microorganism	MIC 50 (µL/mL)	MIC 90 (µL/mL)	MIC 50 (µL/mL)	MIC 90 (µL/mL)
	*T. mastichina EO*		*E. cardamomum EO*	
*Escherichia coli* CCM 3954	3.56	5.75	49.86	56.21
*Haemophilus influenzae* CCM 4454	0.86	2.11	1.65	3.48
*Streptococcus pneumoniae* CCM 4501	12.35	15.46	49.86	56.21
*Listeria monocytogenes* CCM 4699	1.65	3.48	49.86	56.21
*Staphylococcus aureus* CCM 2461	12.35	15.46	49.86	56.21
*Yersinia enterocolitica* CCM 7204	0.86	2.11	1.65	3.48
*Pseudomonas fluorescens*-biofilm	6.36	9.54	0.86	2.11
*Candida glabrata* CCM 8270	0.86	2.11	0.86	2.11
*Candida tropicalis* CCM 8264	6.36	9.54	1.65	3.48
*Candida albicans* CCM 8261	12.35	15.46	12.35	15.46

## Data Availability

Not applicable.

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
