# Peer review of "GC, GC/MS Analysis, and Biological Effects of Essential Oils from Thymus mastchina and Elettaria cardamomum"

_plants, 2022, doi:10.3390/plants11233213_

Round 1

Reviewer 1 Report

The manuscript is based on a large amount of scientific data derived from carefully undertaken scientific work. The presentation is clear for the most part but does require some editing of language. More specific points to be considered during revision follow.

Abstract – EO must be defined, and again when first used in the main body of the manuscript; similarly for genera of microorganisms.

Introduction – reference [1] is not used in the text; there is a substantial amount of text in the Discussion section that would more suitably be placed in the Introduction.

Figure 1 – why are the bars for the control, i.e., 0 µg/mL not shown?

In Figures 1-3 – statistics related to differences between treatments and the control are presented - why were statistics related to differences between treatments (other than error bars) not presented?

Lines 154-156 – this sentence needs to be written more clearly.

Lines 164-165 – this sentence could be deleted; if retained, “and” should be changed to “or”.

Lines 170 – 171 – statistical treatment is necessary to make this statement.

Line 183 – start a new paragraph with the sentence beginning in this line.

Lines 210-221 – the zone inhibition values and MICs don’t seem to correlate all that well; is there an explanation for this?

Lines 216-217 – what is meant by “resp.”?

Line 245, lines 247-248 – why is data for planktonic cells presented rather than data for untreated biofilm?

Line 246 – there is very little narrative comparing biofilms grown on plastic and stainless steel, particularly in section 2.7.2.

Figures 5 and 7 – comparisons between treatments are difficult due to the large number of peaks in the spectrograms – the dendrograms are helpful, but is there a way to assist readers in visualizing differences by making reference to particular peaks in the figures, for example?

Lines 334-335 – this should become part of the subsequent paragraph.

Line 372 – ROS needs to be defined.

Sections 4.10, 4.11 and 4.12 – references should be provided for the methodology in these sections.

Author Response

Dear Reviewer

Thank you for the review of our manuscript entitled ‘GC, GC/MS analysis and biological effects of essential oils from Thymus mastchina and Elettaria cardamomum’. The Authors are very grateful to the Reviewer for their valuable comments. We would like to thank the Reviewer for the time devoted for constructive and important comments to improve our paper. All possible changes in the manuscript have been introduced.

Yours sincerely,

Miroslava Kačániová

Reviewer #3

The manuscript is based on a large amount of scientific data derived from carefully undertaken scientific work. The presentation is clear for the most part but does require some editing of language. More specific points to be considered during revision follow.

  1. Abstract – EO must be defined, and again when first used in the main body of the manuscript; similarly for genera of microorganisms.

Response:

The abbreviations have been defined.

  1. Introduction – reference [1] is not used in the text; there is a substantial amount of text in the

Discussion section that would more suitably be placed in the Introduction.

Response:

Reference [1] has been added to the introduction.

  1. Figure 1 – why are the bars for the control, i.e., 0 µg/mL not shown?

Response:

The control bar has been added in Figure 1.

  1. In Figures 1-3 – statistics related to differences between treatments and the control are presented - why were statistics related to differences between treatments (other than error bars) not presented?

Response:

Thank you for your comment. Serial increasing concentrations of EOs were tested compared to non-treated cells. In all Figures the comparisons were made only between control and doses of treatments, no specific groups of doses and their biological effects were established, which is why no comparisons between individual doses were needed. 

  1. Lines 154-156 – this sentence needs to be written more clearly

(Obtained results showed that all applied treatments of T. mastichina EO after 24 h caused an increase in cell viability, especially in concentrations (1, 10, and 20 µg/mL), where the cell viability was significantly increased compared to control cells, while in concentration 100 and 200 µg/mL cell viability was reduced compared to control, but they were not statistically significant.)

Response:

The sentence was rewritten according to your suggestions.

  1. Lines 164-165 – this sentence could be deleted; if retained, “and” should be changed to “or”.

(In order to obtain more comprehensive data on the influence of the examined treatments on the redox homeostasis of breast cancer, as established indicators of oxidative stress, the concentrations of the following parameters were determined: O2•−, NO, and total GSH.)

Response:

Thank you for the suggestion. The sentence is shortened, cleared, and rewritten accordingly.

  1. Lines 170 – 171 – statistical treatment is necessary to make this statement.

(The results show that the production of O2•− was decreased compared to the control after short-time exposure to all concentrations.)

Response:

Thank you for your comment. The explanations of this parameter were not precise enough. The sentences are clarified and rewritten accordingly.

  1. Line 183 – start a new paragraph with the sentence beginning in this line.

Response:

Thank you for the comment the sentence has been added to the paragraph.

  1. Lines 210-221 – the zone inhibition values and MICs don’t seem to correlate all that well; is there an explanation for this?

Response:

It is expected for these two methods to correlate, but as there are differences in conditions for the growth of microorganisms (solid/liquid agar, concentration used for MIC) small variations in the result are possible. Still, both results show good antimicrobial activity of essential oils.

  1. Lines 216-217 – what is meant by “resp.”?

Response:

An acronym has been defined.

11.Line 245, lines 247-248 – why is data for planktonic cells presented rather than data for untreated biofilm?

Response:

An explanation has been added to the article. “The spectra of biofilms and planktonic cells in the control group developed identically and, therefore, the spectra of planktonic cells were used for greater clarity than the control spectrum.”

  1. Line 246 – there is very little narrative comparing biofilms grown on plastic and stainless steel, particularly in section 2.7.2.

Response:

I assume from your question that it was section 7.2.1 as the changes were described at length in section 7.2.2. Added to section 7.2.1 was "Differences in protein profile evolution between plastic surface and stainless steel were not noted."

  1. Figures 5 and 7 – comparisons between treatments are difficult due to the large number of peaks in the spectrograms – the dendrograms are helpful, but is there a way to assist readers in visualizing differences by making reference to particular peaks in the figures, for example?

Response:

We will consider such clarification in the future. But in our opinion, it is easiest for the reader to see the change in the shape of the spectrum as a change in the values of the individual peaks.

  1. Lines 334-335 – this should become part of the subsequent paragraph

Response:

The paragraphs were linked.

  1. Line 372 – ROS needs to be defined.

Response:

An acronym has been defined.

  1. Sections 4.10, 4.11 and 4.12 – references should be provided for the methodology in these sections.

Response:

References have been added.

Reviewer 2 Report

Overall the manuscript is ok. I have a few questions.

1. How the author calculated the diameter of the zone of inhibition as 5 or less mm, as they have used a 6mm size of sterile discs?

2. which solvent is used to dilute the essential oil for the assays?

3. Author should carry out statistical analysis for the antimicrobial results.

4. Author should provide images of the disc diffusion assay.

Author Response

Reviewer #1

Overall, the manuscript is ok. I have a few questions.

The Authors are very grateful to the Reviewer for their valuable comments. We would like to thank the Reviewer for the time devoted for constructive and important comments to improve our paper.

Point 1: How the author calculated the diameter of the zone of inhibition as 5 or less mm, as they have used a 6mm size of sterile discs?

Response: A more precise description of the inhibition zone radius measurement has been added to the disk diffusion method. The radius of the inhibition zone was measured from the disk to the edge of the zone.

Point 2: which solvent is used to dilute the essential oil for the assays?

Response: For minimum inhibitory concentration analysis, essential oils were diluted in Mueller Hinton Broth and Sabouraud Dextrose Broth depending on whether they were bacteria or yeast.

Point 3: Author should carry out statistical analysis for the antimicrobial results.

Response: The antimicrobial results were supplemented One-way analysis of variance (ANOVA) was performed using Prism 8.0.1 (GraphPad Software, San Diego, CA, USA), followed by Tukey’s test at p < 0.05.

Point 4: Author should provide images of the disc diffusion assay.

Response: We are sorry but we are not used to taking photographs for the disk diffusion method and therefore cannot document them. In the future, we will also perform photo documentation of inhibition zones.

Reviewer 3 Report

The work was good in presentation and sounds promising but it still has some errors in typing. The authors need to recheck again all the manuscript

I still have some questions to ask

- Why did the authors work out with the disc diffusion method instead of the viable assay  with tetrazolium chloride and determined it by ELISA Plate

-  The two plants were discussed before by different authors why the authors chose them 

- Why the authors did not analyze their data by any statical test like HSD post hoc test

- "Results and discussion" section name should be corrected to Results

- I wonder why the authors made a GC-MS analysis for the two essential oils which were bought from the company. What I know is that if you hydrodistilled the EO by yourself you can go to characterize it ???

- Where is the accession number of the Pseudomonas isolate you used in your study ????

Regards

Author Response

Reviewer #2

The work was good in presentation and sounds promising, but it still has some errors in typing. The authors need to recheck again all the manuscript.

I still have some questions to ask.

The Authors are very grateful to the Reviewer for their valuable comments. We would like to thank the Reviewer for the time devoted for constructive and important comments to improve our paper.

Point 1: Why did the authors work out with the disc diffusion method instead of the viable assay with tetrazolium chloride and determined it by ELISA Plate

Response: MTT assay was used to determine cell viability. The disc diffusion method and the minimum inhibitory concentration method were used in the next steps to determine the antimicrobial activity of the essential oils analyzed.

Point 2: The two plants were discussed before by different authors why the authors chose them.

Response: The essential oils analyzed by us were chosen also because of their positive properties published by other authors, but we analyzed essential oils commercially purchased from Hanus s.r.o. And as it is known, the effectiveness of essential oils also depends on their origin, the way they are obtained, and the growing conditions of the parent plant. We aimed to verify the effectiveness of essential oils from this producer.

Point 3: Why the authors did not analyze their data by any statical test like HSD post hoc test

Response: It was supplemented One-way analysis of variance (ANOVA) was performed using Prism 8.0.1 (GraphPad Software, San Diego, CA, USA), followed by Tukey’s test at p < 0.05.

Point 4: "Results and discussion" section name should be corrected to Results.

Response: Thank you for your comment. The name of the section has been changed.

Point 5: I wonder why the authors made a GC-MS analysis for the two essential oils which were bought from the company. What I know is that if you hydrodistilled the EO by yourself you can go to characterize it???

Response: Yes, we have performed a GC-MS analysis as the manufacturer only gives very strict data on the composition of the essential oils.

Point 6: Where is the accession number of the Pseudomonas isolate you used in your study????

Response: Pseudomonas fluorescens for antibiofilm activity has not been obtained from a collection of microorganisms therefore it does not have a numerical designation. Biofilm-producing strains are not commonly available in collections. The P. fluorescens we used was isolated from fish samples in previous studies.

Round 2

Reviewer 2 Report

The author has made the necessary revision.

Author Response

Many thanks for your comments.

Reviewer 3 Report

The manuscript could be accepted for publishing in the journal 

Author Response

Many thanks for your comments.